# Acceptance of COVID-19 Vaccine among the Healthcare Workers in the Eastern Cape, South Africa: A Cross Sectional Study

**DOI:** 10.3390/vaccines9060666

**Published:** 2021-06-18

**Authors:** Oladele Vincent Adeniyi, David Stead, Mandisa Singata-Madliki, Joanne Batting, Matthew Wright, Eloise Jelliman, Shareef Abrahams, Andrew Parrish

**Affiliations:** 1Department of Family Medicine and Rural Health, Walter Sisulu University, Cecilia Makiwane Hospital, East London Hospital Complex, East London 5219, South Africa; 2Department of Internal Medicine, Cecilia Makiwane and Frere Hospitals, Walter Sisulu University, East London 5219, South Africa; dfstead@gmail.com (D.S.); mattymoowright@gmail.com (M.W.); andygp@mweb.co.za (A.P.); 3Effective Care Research Unit, Department of Obstetrics & Gynaecology, Frere Hospital, University of Fort Hare, University of Witwatersrand, East London 5200, South Africa; mandisa.singata@gmail.com (M.S.-M.); joannebatting@hotmail.com (J.B.); 4Department of Radiology, Frere Hospital, Walter Sisulu University, East London 5200, South Africa; jellimaneloise@gmail.com; 5Department of Pathology, Division of Medical Microbiology, National Health Laboratory Service, Port Elizabeth 6001, South Africa; shareef.abrahams@nhls.ac.za

**Keywords:** Eastern Cape, healthcare workers, SARS-CoV-2, South Africa, vaccine acceptance

## Abstract

Background: This study assesses the perceptions and acceptance of severe acute respiratory syndrome coronavirus-2 (SARS-CoV-2) vaccination. It also examines its influencing factors among the healthcare workers (HCWs) in the Eastern Cape, South Africa. Methods: In this cross-sectional study performed in November and December 2020, a total of 1308 HCWs from two large academic hospitals participated in the Eastern Cape Healthcare Workers Acquisition of SARS-CoV-2 (ECHAS) study. Validated measures of vaccine hesitancy were explored using a questionnaire. Logistic regression was used to identify the determinants of vaccine hesitancy. Results: The majority were nurses (45.2%), and at risk for unfavourable Covid-19 outcome, due to obesity (62.9%) and having direct contact with individuals confirmed to have Covid-19 (77.1%). The overall acceptance of SARS-CoV-2 vaccine was 90.1%, which differed significantly by level of education. Individuals with lower educational attainment (primary and secondary education) and those with prior vaccine refusal were less likely to accept the SARS-CoV-2 vaccine. However, positive perceptions about the SARS-CoV-2 vaccine were independently associated with vaccine acceptance. Conclusions: The high level of acceptance of SARS-CoV-2 vaccine is reassuring; however, HCWs with a lower level of education and those with prior vaccine refusal should be targeted for further engagements to address their concerns and fears.

## 1. Introduction

Since the identification of the severe acute respiratory syndrome coronavirus-2 (SARS-CoV-2) causing Coronavirus disease 2019 (COVID-19) and the global pandemic, multiple groups began the race to produce safe and effective vaccines. To date, several vaccines have received approval by the regulating authorities and are being rolled-out in most countries (including South Arica) as a strategy to end the global pandemic. While experts agree that large scale vaccination of populations is the best strategy to gain control of the pandemic through sufficient ‘herd immunity’, convincing citizens that vaccines are safe, effective and necessary is an essential component to any effective vaccination programme.

Vaccine hesitancy refers to delay in acceptance or refusal of vaccination despite its availability, and has been encountered since the invention of vaccines in 1796 by Edward Jenner [1]. Alarmingly, this has not diminished with growing scientific sophistication and improved education. The World Health Organisation strategic advisory group of experts (SAGE) on vaccines, produced a report on vaccine hesitancy in 2014. It categorised reasons for vaccine hesitancy into three groups: confidence (trust in healthcare professionals, vaccines, and their effectiveness), complacency (low awareness of the risks of vaccine preventable diseases and the importance of vaccine) and convenience (availability of and accessibility to vaccines and healthcare services) [2]. The role of healthcare workers (HCWs) in instilling confidence in the general population about vaccine safety and efficacy has been documented in previous studies [2,3]. Evidence suggests a strong link between vaccine hesitancy among HCWs and the general population. The HCWs with positive attitudes to, and more knowledge regarding vaccines, produce higher vaccination rates in their patients [4].

In a recent global online poll of around 20,000 people by the Institut de Publique Sondage d’Opinion Secteur (IPSOS), 74% of adults surveyed prior to SARS-CoV-2 vaccine rollouts agreed to get vaccinated. Out of the South Africans surveyed (approximately 500 participants), only 64% reported that they would agree to get vaccinated against SARS-CoV-2, being the fifth lowest vaccine acceptance of the 27 countries sampled [5]. Of concern is that the reported acceptance of the SARS-CoV-2 vaccine among HCWs was largely below this global average; ranging from 27% in the Democratic Republic of Congo to 75% in France [6,7,8,9,10]. Reasons for SARS-CoV-2 vaccine hesitancy among HCWs included concerns regarding the expedited vaccine development, political interference, current political climates, and concerns regarding serious side effects despite proven safety and efficacy [11]. Certain demographic factors were found to be associated with higher rates of vaccine acceptance among HCWs: male sex, older age, higher level of education, physicians, pharmacists, frontline HCWs and having previous vaccination for seasonal influenza. In addition, perceived individual vulnerability to SARS-CoV-2 due to co-morbidities and/or high virus exposure within the workplace positively influenced vaccine acceptance [7,10,12].

South Africa has suffered through two waves of COVID-19 with over 1.5 million people infected, and over 40,000 recorded deaths to date. Health care workers have suffered high infection rates, especially in the Eastern Cape province, with 11,262 HCWs infected and 262 deaths by the 18 February 2021 [13]. Getting good buy-in and close to 100% vaccine coverage of the HCW populations is imperative in the region. Beside the promise of high HCW vaccination uptake reversing the negative impact of COVID-19 disease on health services, its potential positive knock-on effect on the success of the mass roll-out of vaccines for the general population makes it compelling. However, there is currently no published data on the perceptions and acceptance of SARS-CoV-2 vaccine among HCWs in South Africa, which this study sought to address. Therefore, this study assesses the risk perceptions, level of acceptance of SARS-CoV-2 vaccine and further, examines the influencing factors of vaccine hesitancy among the HCWs in the Eastern Cape, South Africa.

## 2. Materials and Methods

### 2.1. Design and Settings

This cross-sectional survey was a component of a larger Eastern Cape Healthcare Workers Acquisition of SARS-CoV-2 (ECHAS) Study, which studied SARS-CoV-2 antibody seroprevalence among HCWs [14]. Two large academic hospitals in the central region of the Eastern Cape Province were studied. Frere hospital provides tertiary care services for four districts with a combined population of almost 3 million people, with almost 20 major departments and a range of sub-specialties [15]. Cecilia Makiwane hospital provides both level one and two services for a combined population of 1.6 million people in two districts. These two hospitals have a combined staff of almost 4000 and are both affiliated to Walter Sisulu University [15].

### 2.2. Participants

Eligible participants were doctors, nurses, allied health workers, administrative and support staff aged 18 years or older that were permanently employed by either of the hospitals. All staff were given an opportunity to participate in the study over a period of six weeks between November and December 2020. This study adopted a multi-stage cluster sampling technique in order to ensure a representative sample of HCWs recruited across the various domains within the hospitals. In order to ensure inclusivity, exposure areas were pre-defined as clusters in accordance with the risk assessment by Iversen et al. as high risk (Accident and Emergency unit, acute respiratory (person under investigation/COVID-19) wards and intensive care units (ICU)); Intermediate risk (non-respiratory admission wards, outpatient departments (OPDs) and other clinical areas) and low risk (administrative offices and other non-clinical areas) [16]. Participants were then conveniently sampled within each cluster, with voluntary participation. In addition, a proportionate sample of the various professional categories was recruited. In order to avoid time variations in the main outcome measure between the two hospitals, participants were recruited concurrently. Four research assistants and two research nurses who were proficient in English and IsiXhosa (local language), were trained for two days and were allocated to a study site.

Each of the HCWs completed a self-administered structured questionnaire (pre-piloted with five HCWs specifically for the study). Few participants received support from the research assistants with the completion of the questionnaire. Data were subsequently captured into Research Electronic Data Capture (Redcap) online software with a pre-installed questionnaire. This electronic software is housed by the South African Medical Research Council Server for privacy and confidentiality of data.

### 2.3. Measures

We obtained relevant demographic data, which included age, sex, race, type of residence, educational level and number of household members. In addition, clinical conditions that have been shown to increase the risks for severe COVID-19 disease or mortality (diabetes mellitus, hypertension, tuberculosis, chronic kidney disease, heart disease and chronic lung disease) were self-reported by the participants [17,18,19]. We hypothesized that the presence of at least one of the co-morbidities would influence the decision to accept Covid-19 vaccination among the HCWs. The research nurses performed anthropometric measurements (height and weight) according to the standard protocols. The body mass index (BMI) was estimated and categorised as obese if BMI ≥ 30.0 kg/m [2,20].

Exposure to SARS-CoV-2 was assessed by documenting the domain of work within the hospital and additional questions on perceived exposure were elicited: “Have you had direct contact with anyone with COVID-19 at hospital?”, “Have you had direct contact with individuals diagnosed with COVID-19 outside of the hospital?” and “Have you ever been diagnosed with COVID-19?”. It was postulated that HCWs who perceived themselves to be highly vulnerable to contract Covid-19 might be more receptive to accept vaccine. This hypothesis was further tested in the models.

### 2.4. Outcomes

Acceptance of SARS-CoV-2 vaccine was the main outcome measure and assessed with the question: “When COVID-19 vaccine becomes available; will you personally receive the vaccine? Responses were categorised as ‘yes’ or ‘no’.

General vaccine attitudes of the HCWs were assessed with a set of questions: “Do you believe that a vaccine is needed to end the COVID-19 pandemic?”, “Do you think every HCW should get the COVID-19 vaccine when it becomes available?” and ‘Do you think vaccines are generally safe?” Responses to these questions were categorised as ‘yes’ or ‘no’ options. The HCWs’ attitudes toward vaccines were also elicited with two sets of questions: “Have you ever refused vaccines in the past?” and “Have you ever experienced adverse effects from vaccines before?”. A “yes” or “no” response option was provided to elicit the underlying attitude of the HCWs to vaccines in general that could impact on their willingness to accept future COVID-19 vaccines.

### 2.5. Ethical Considerations

The Walter Sisulu University Faculty of Health Sciences Ethics Committee granted approval for the implementation of the study protocol (Project Identification Code: 087/2020). Permission was obtained from the Eastern Cape Department of Health as well as the clinical governance of the two hospitals. Information sharing sessions about the study were disseminated through the clinical managers, nursing managers and union representatives. In addition, participants received group information as well as an information sheet detailing the purpose and process of the study prior to signing an informed consent indicating their voluntary participation. Participants’ right to privacy and confidentiality of medical information was respected during and after the study. Unique patient identifying numbers (PTID) were used for coding in order to ensure privacy of medical information. The study was conducted in accordance with the Helsinki Declaration and Good Clinical Practice Guidelines and principles governing human research.

### 2.6. Validity and Reliability of the Study Instruments

The construct and criterion validity of the instrument have been established by selecting variables that have been used successfully in measuring the outcome measures of this study in previous studies [18,19,21,22,23,24]. In addition, the instrument was pre-tested with five HCWs at one of the study sites and the feedback from the participants was critically reviewed by the investigators, following which adjustments were made to the tool. The results of the pre-test were not included in the main study.

### 2.7. Statistical Analysis

Complete data were captured on Redcap and analysed by using the IBM SPSS Statistics for Windows, Version 27.0 (IBM Corp., Armonk, NY, USA). All data entry errors were corrected after running a simple frequency for all the variables. The baseline characteristics of the study participants are presented as categorical variables with frequencies and percentages.

The relationship between the main outcome measure (acceptance of Covid-19 vaccine) and participants’ characteristics (baseline characteristics and perception about Covid-19 vaccine) were assessed using chi-square analysis. The direction of the association was analysed using bivariate logistic regression analysis (stepwise forward L-R Method) with a 95% confidence interval (95% CI). Selection of variables into the bivariate logistic regression model was based on factors that were previously reported in the literature and found to be significant in the chi-square analysis. Using a stepwise forwards L-R method, level of education was selected as the most influential factor. Therefore, the final model was composed of the following variables: “Level of education”, “Is the vaccine needed to end the pandemic?”, “Should healthcare workers receive the vaccine?”, “Are vaccines safe?”, “Past vaccine refusal?” and “Experienced adverse reactions from a vaccine”. A *p*-value of less than 0.05 was considered statistically significant.

## 3. Results

### 3.1. Baseline Characteristics of the Participants

Of the total participants (N = 1308) included in the study, nursing staff were the majority (45.2%), followed by support staff (28.7%) and pharmacy staff accounted for the lowest proportion (4.7%). The two hospitals contributed almost equal number of participants (ratio 1:1). The majority of the participants were females (81.5%), aged 26–55 years (79.1%), obese (62.9%) and were of black ethnicity (78.8%). Most of the participants reside in urban areas (90.1%), have attained a tertiary education (71.4%), have never smoked cigarettes (91.1%), have at least one co-morbidity (65.6%) and had not at the time been diagnosed with Covid-19 (69.7%) (Table 1).

### 3.2. Acceptance of SARS-CoV-2 Vaccine and Their Associations with Baseline Characteristics

The overall acceptance of SARS-CoV-2 vaccine was 90.1%, with significant difference by level of education (*p* < 0.001) and professional category (*p* = 0.021). However, there was no significant difference in the acceptance of Covid-19 vaccine by sex, BMI categories, race, smoking status, presence of co-morbidities and prior diagnosis of Covid-19 (*p* > 0.05) (Table 1).

### 3.3. Perception of Risks, SARS-CoV-2 Vaccine and Their Associations with Vaccine Acceptance

A large proportion of the participants (77.1%) reported direct contact with individuals diagnosed with Covid-19 within the hospital while fewer participants (26.5%) confirmed direct COVID-19 contact outside of the hospital. The majority of the participants believed that vaccine will be needed to end the pandemic (90.1%) and that HCWs should receive the vaccine (92.7%). A slightly lower proportion considered vaccines to be safe (86.9%) and had not refused vaccine in the past (86.5%). Most had not experienced adverse events from previously administered vaccines (90.2%). There were no significant associations between perceived risk of exposure (direct contact with individuals with Covid-19 within or outside the hospital) and vaccine acceptance. However, all of the attitude questions regarding vaccines (vaccines are needed to end the pandemic, HCWs should receive the vaccines, vaccines are safe, acceptance of vaccines in the past and no prior experience of adverse events following vaccination) were significantly associated with vaccine acceptance (Table 2).

### 3.4. Predictors of Covid-19 Acceptance among HCWs

In the bivariate logistic regression analysis, individuals with primary (OR 0.14 95% CI 0.33–0.64) or secondary education (OR 0.06, 95% CI 0.01–0.33) were less likely to accept the vaccine in comparison to those with tertiary education. Participants who had refused a vaccine in the past were less likely to accept the vaccine (OR 0.52, 95% CI 0.28–0.99) in comparison to those who had not.

However, positive attitudes toward the SARS-CoV-2 vaccine were all significantly associated with vaccine acceptance, including belief that vaccines are needed to end the pandemic (OR 3.70, 95% CI 1.99–6.89), belief that vaccines are safe (OR 9.48, 95% CI 5.67–15.84) and belief that vaccine should be given to every HCW (OR 18.67, 95% CI 9.78–35.64) (Table 3).

## 4. Discussion

There is consensus among experts that large scale vaccination against SARS-CoV-2, leading to attainment of population level “herd immunity” is the best strategy to control the pandemic. However, hesitancy against SARS-CoV-2 vaccine is mounting globally. Therefore, it is essential to convince people at the community level that vaccines are safe, effective and necessary. Health care workers are uniquely positioned to lend their voices and actions in promoting this biomedical strategy to their patients and the community at large, especially in South Africa, which has experienced significant devastation from the COVID-19 pandemic. However, there is currently no published data on the perceptions and acceptance of SARS-CoV-2 vaccine among HCWs in South Africa. This study reports the perceptions and level of acceptance of SARS-CoV-2 vaccine, and further examines the influencing factors of vaccine acceptance among HCWs in the Eastern Cape, South Africa.

This survey of 1308 hospital-based HCWs revealed a reassuringly high level of acceptance of the SARS-CoV-2 vaccine (90.1%) in the region. This is higher than previous reports of HCWs’ vaccine acceptance rates of 27% to 75% in other countries, and the general South African public survey of 64% [5,6,7,8,10]. Due to the high risk of HCWs contracting SARS-CoV-2 infection, a 100% vaccine coverage of staff would be desirable, hence, an understanding of factors influencing vaccine acceptance is important. Vaccine acceptance was significantly higher in HCWs with tertiary education in this study. While there was no significant difference between any professional category of workers, education level was a major factor. Many studies have demonstrated a dose-response effect between the level of educational attainment and vaccine acceptance [25,26,27]. Lower educated categories of HCWs could benefit from targeted vaccine education efforts.

Factors such as older age, raised BMI, presence of co-morbidities and prior diagnosis of Covid-19, which may influence an individual’s perceived risk of COVID-19 (and hence need for vaccination) were not associated with vaccine acceptance in this study. This is in contrast to other studies that have reported higher vaccine acceptance with HCWs who were male, of older age and with co-morbid illnesses [7,10,12]. Unsurprisingly, HCWs’ overall vaccine beliefs significantly correlated with their willingness to accept SARS-CoV-2 vaccination. The belief that vaccines are needed to end the SARS-CoV-2 pandemic reflects a broader understanding that vaccination is important on a population level. The belief that all HCWs should receive the vaccine reflects an understanding of the importance of protecting HCWs as a high-risk group. Individuals’ confidence in vaccinations in general was measured by asking whether vaccines were considered generally safe, which also correlated with SARS-CoV-2 vaccine acceptance.

This study was performed between November and December 2020, when the first SARS-CoV-2 vaccine trial results had been released (Pfizer and Moderna initially), and these vaccines were being registered in a few countries. Hence, these results need to be interpreted in the context of the SARS-CoV-2 vaccine information that was available at that time. With the rapid vaccine development and deployment globally to date, as well as subsequent concerns raised by regulatory bodies over rare but severe adverse events such as vaccine-induced immune thrombotic thrombocytopaenia (VITT), an individual’s perception of vaccine safety may change daily [28]. More so, there is emerging evidence that the South African variants of SARS-CoV-2 (B.1.351) have some resistance to neutralisation by convalescent plasma and certain vaccines [29,30]. This information may further erode the confidence in vaccines and thus, negatively impact the level of acceptance of the SARS-CoV-2 vaccines at the population level. Future studies could potentially explore how the genetic variants of SARS-CoV-2 influence the level of acceptance of the vaccines against the virus.

A potential general distrust in vaccines was assessed by reporting refusal of vaccination in the past, which was associated with SARS-CoV-2 vaccine refusal. Vaccines routinely offered to adult HCWs in the South African state sector are Hepatitis B and an annual influenza vaccination, but these are voluntary. Having experienced adverse events from vaccines in the past also correlated with vaccine refusal in this study. These two vaccine questions gave some insight into the reasons for HCWs’ vaccine refusal in this context. Rather than specific concerns with the SARS-CoV-2 vaccine safety, efficacy or need, this sub-group appears to have a general distrust or fear of vaccines, partly informed by past vaccine adverse events. Adverse effects of vaccines have been reported previously to lead to vaccine refusal [31,32].

An individual vaccine acceptance decision represents a complex sum of vaccine confidence, disease complacency (or fear) and convenience to access vaccination (as per the WHO advisory group) [2]. The first two are strongly rooted in an individual’s cultural and religious beliefs, and informed by their level of education, and access to scientifically sound information. South Africa has a strong emphasis on human rights, and the health minister has emphasised that SARS-CoV-2 vaccination will be entirely voluntary. However, the success of the country’s ambitious vaccination programme, depends on high levels of uptake of the vaccine by eligible citizens to achieve herd immunity. This is particularly crucial among HCWs, who are both at high risk of infection, as well as being an essential work-force in the COVID-19 response. Positive attitudes toward vaccination amongst HCWs will result in higher uptake amongst the general public. While the high levels of acceptance in this survey were encouraging, the almost 10% who did not plan to get vaccinated need to be engaged and their concerns addressed for the success of the national programme.

### Study Limitations

These findings represent a snapshot of the perceptions, attitudes and acceptance of SARS-CoV-2 vaccine at the time of the study, given that these phenomena are dynamic in nature and may be influenced easily by safety and efficacy data. In addition, the binary nature (yes/no) of the main outcome (vaccine acceptance) did not allow for better understanding of the timing of acceptance of the vaccine. While the majority of the staff will probably accept the vaccine immediately whenever it becomes available others might choose to defer till more people have received the vaccine in the country. Perhaps, a five-point Likert scale showing varying degrees of opinions on vaccine acceptance would have shed more light on the overall perceptions and attitudes of the staff. Though, the investigators provided extended access to all eligible participants, the potential bias due to the convenience sampling and voluntary participation of the HCWs cannot be ignored. Notwithstanding the limitations, the findings are reassuring that HCWs were willing to accept and recommend the vaccine to other HCWs. The staff who chose not to accept the vaccine need further engagement targeting their fears about adverse events and myths surrounding the SARS-CoV-2 vaccine. It should also be noted that this study was implemented in November and December 2020 prior to the roll-out of the SARS-CoV-2 vaccine in South Africa. Therefore, uptake of the SARS-CoV-2 vaccine by the HCWs should be monitored in light of the findings reported in this study.

## 5. Conclusions

The high level of acceptance of SARS-CoV-2 vaccine among the HCWs (90.1%) in the region is reassuring. Findings that healthcare workers were willing to accept and believed that others should receive SARS-CoV-2 vaccine highlight a positive signal towards the success of a mass vaccination programme in the region. However, healthcare workers with lower levels of education and those with prior vaccine refusal should be targeted for further engagements to address their fears and concerns. It will be crucial to monitor the uptake of the SARS-CoV-2 vaccines whenever they become available and correlate the refusal categories with the findings in this study.

## Figures and Tables

**Table 1 vaccines-09-00666-t001:** Pearson chi-squared test of demographic characteristics of the participants and SARS-CoV-2 vaccine acceptance.

Variables	Total	Acceptance of SARS-CoV-2 Vaccine		*p*-Value
		Yes	No	
Sex	N = 1308			0.25
Male	242 (18.5)	223 (92.2)	19 (7.9)	
Female	1066 (81.5)	956 (89.7)	110 (10.3)	
Age (Years)				
18–25	74 (5.7)	65 (87.8)	09 (12.2)	0.55
26–35	328 (25.1)	293 (89.3)	35 (10.7)	
36–45	353 (27.0)	326 (92.4)	27 (7.6)	
46–55	354 (27.1)	318 (89.8)	36 (10.2)	
>55	204 (15.6)	182 (89.2)	22 (10.8)	
* BMI (Kg/m^2^)				0.24
<18.5	07 (0.5)	06 (85.7)	01 (14.3)	
18.5–24.9	179 (13.8)	154 (86.0)	25 (14.0)	
25.0–29.9	297 (22.8)	269 (90.6)	28 (9.4)	
≥30	818 (62.9)	744 (91.0)	74 (9.1)	
Race				0.06
Black	1030 (78.8)	940 (91.3)	90 (8.7)	
White	115 (8.8)	99 (86.1)	16 (13.9)	
Coloured	34 (2.6)	17 (50.0)	17 (50.0)	
Others	53 (4.1)	47 (88.7)	06 (11.3)	
Level of education				<0.001
Primary	13 (1.0)	08 (61.5)	05 (38.5)	
Secondary	361 (27.6)	344 (95.3)	17 (4.7)	
Tertiary	934 (71.4)	827 (88.5)	107 (11.5)	
Facility				0.10
FH	659 (50.4)	585 (88.8)	74 (11.2)	
CMH	649 (49.6)	594 (91.5)	55 (8.5)	
Profession				0.02
Doctors	176 (13.5)	158 (89.8)	18(10.2)	
Pharmacy Staff	61 (4.7)	58 (95.0)	03 (4.9)	
Nurses	591 (45.2)	527 (89.2)	64 (10.8)	
Allied Health Staff	106 (8.1)	88(83.0)	18 (17.0)	
Support Staff	375 (28.7)	348 (92.8)	27 (7.2)	
Presence of at least one co-morbidity				0.21
Yes	858 (65.6)	767 (89.4)	91 (10.6)	
No	450 (34.4)	412 (91.6)	38 (8.4)	
Previous Covid-19 Diagnosis				0.27
Positive	401 (30.7)	356 (88.8)	45 (11.2)	
Negative	912 (69.7)	828 (90.8)	84 (9.2)	
Direct contact with anyone diagnosed with Covid-19 at the hospital				0.68
Yes	1009 (77.1)	906 (89.8)	103 (10.2)	
No	297 (22.7)	27 (91.3)	26 (8.8)	
Direct contact with anyone diagnosed with Covid-19 outside hospital				0.42
Yes	346 (26.5)	308 (89.0)	38 (11.0)	
No	962 (73.6)	871 (90.5)	91 (9.5)	

BMI = Body mass index; CMH = Cecilia Makiwane Hospital; Covid-19 = Coronavirus disease 2019; FH = Frere Hospital.

**Table 2 vaccines-09-00666-t002:** Perception of risks, SARS-CoV-2 vaccine and their associations with acceptance.

Variables	Total *n* (%)	Acceptance of Covid-19 Vaccine		*p*-Value
		Yes	No	
Direct contact with anyone diagnosed with Covid-19 at the hospital				0.68
Yes	1009 (77.1)	906 (89.8)	103 (10.2)	
No	297 (22.7)	27 (91.3)	26 (8.8)	
Direct contact with anyone diagnosed with Covid-19 outside hospital				0.42
Yes	346 (26.5)	308 (89.0)	38 (11.0)	
No	962 (73.6)	871 (90.5)	91 (9.5)	
Is the vaccine needed to end the pandemic?				<0.001
Yes	1178 (90.1)	1112 (94.4)	66 (5.6)	
No	130 (9.9)	67 (51.5)	63 (48.5)	
Should healthcare workers receive the vaccine?				<0.001
Yes	1213 (92.7)	1153 (95.1)	60 (5.0)	
No	95 (7.3)	26 (27.4)	69 (72.6)	
Are vaccines safe?				<0.001
Yes	1137 (86.9)	1085 (95.4)	52 (4.6)	
No	171 (13.1)	94 (55.0)	77 (45.0)	
Past vaccine refusal				0.01
Yes	177 (13.5)	150 (84.8)	27 (15.3)	
No	1131 (86.5)	1029 (91.0)	102 (9.0)	
Experienced adverse reaction from a vaccine				0.02
Yes	128 (9.8)	108 (84.4)	20 (15.6)	
No	1180 (90.2)	1071 (90.8)	109(9.2)	

Covid-19 = Coronavirus disease 2019.

**Table 3 vaccines-09-00666-t003:** Binary logistic regression analysis of factors associated with vaccine acceptance.

Variable	Odds Ratio (95% CI)	*p*-Value
Education		
Tertiary	Ref	
Secondary	0.06(0.01–0.33)	<0.001
Primary	0.14(0.33–0.64)	0.01
Is the vaccine needed to end the pandemic?		
No	Ref	
Yes	3.70(1.99–6.89)	<0.001
Should healthcare workers receive the vaccine?		
No	Ref	
Yes	18.67(9.78–35.64)	<0.001
Are vaccines safe?		
No	Ref	
Yes	9.48(5.67–15.84)	<0.001
Past vaccine refusal		
No	Ref	
Yes	0.52(0.28–0.99)	0.048

## Data Availability

Data from the ECHAS study is available with the lead investigator (O.V.A.) and will be made available upon reasonable request.

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
