# Peer review of "Acceptance of COVID-19 Vaccine among the Healthcare Workers in the Eastern Cape, South Africa: A Cross Sectional Study"

_vaccines, 2021, doi:10.3390/vaccines9060666_

Round 1

Reviewer 1 Report

In their manuscript, Adeniyi et. al. describe a study of risk perceptions, acceptance, and hesitancy in regards to the SARS-CoV-2 vaccine in South Africa.  

  • The manuscript needs some editing for English language, grammar, syntax, and tense
  • Line 40: Please define COVID-19 at first usage
  • In regards to the hospitals themselves, what percentage of the total HCW worker population does this represent for SA? Are these a representative sample of all SA or just within distinct groups?
  • Convenience sampling and voluntary participation are both potential sources of bias. Was the influence of this aspect of the study design investigated for any sort of skewing of the data?
  • Was there any incentive to participate in the study?
  • Line 131: What standards are the BMI categories chosen by? Please give a reference for this cutoff as these factors can vary by country and region.
  • Section 2.5: Please give specific referral to ethical approval
  • Lines 192-194: The authors state that they attempted to ensure inclusivity but gained a majority of nursing staff in their sampling. Can this be further discussed?
  • Is the data for educational level biased as the study authors are comparing very few primary level individuals with tertiary? This holds true for the bivariate analysis as well. Would a doctor or nurse not have a higher education level but also be more likely to accept a vaccine? Please consider further multivariate analyses or further defining the information in the paper.
  • Lines 270-277: Maybe also good to mention genetic variants of the virus causing issues with vaccine effectiveness and how that can change perception – indeed, this seems to have happened in South Africa due to the February study on AstaZeneca vaccines.
  • The authors mention political sensitivity in their introduction; however, there was no variability to measure how hesitancy or refusal would factor in if governmental trust was compromised. Indeed, this has been a major issue in many countries in the world as they have vaccines available, but the population refuses to be vaccinated due to political distrust (ie: Hong Kong)…please consider touching on this point in the paper.
  • Was any attempt made to measure what type or provenance of the vaccine would influence acceptance? Indeed, this would be a very interesting piece of information whether novel vaccine technology or place of production would influence even highly educated health care workers decisions to be vaccinated.
  • Unfortunately, South Africa has had a very slow vaccination rollout compared to some other countries in their world due to several factors. How many HCWs have actually been vaccinated to date and has their vaccine acceptance been in line with the study results?

Author Response

Comment 1: The manuscript needs some editing for English language, grammar, syntax, and tense.

Response: English editing has been carried out.

Comment 2: Line 40: Please define COVID-19 at first usage

Response: We made correction as suggested.

Comment 3: In regards to the hospitals themselves, what percentage of the total HCW worker population does this represent for SA? Are these a representative sample of all SA or just within distinct groups?

Response: Our study sample is representative of the HCWs working in the two hospitals. The participants’ distribution is proportional to the staff distribution in the two hospitals. Nurses have the highest proportions of the HCWs at these levels of care in the country. Hence, the larger proportion of nurses in our study is consistent with population of HCWs at these institutions.

Comment 4: Convenience sampling and voluntary participation are both potential sources of bias. Was the influence of this aspect of the study design investigated for any sort of skewing of the data?

Response: Thanks for the insightful comment. The investigators encouraged inclusivity by ensuring that every department, professional categories and work areas within the institutions were covered. Also, a central recruitment area was provided to improve access to all the staff. However, the minimal effect of convenience sampling and voluntary bias cannot be ignored. Hence, we have noted this in the limitation of the study.

Comment 5: Was there any incentive to participate in the study?

Response: The investigators noted that HCWs were eager to participate in the study because they wanted to know if they had developed immunity to the virus. However, there were no financial or material incentives to any HCW to participate in the study.

Comment 6: Line 131: What standards are the BMI categories chosen by? Please give a reference for this cut-off as these factors can vary by country and region.

Response: We have provided reference for this categorisation (Project Identification Code: 087/2020).

Comment 7: Section 2.5: Please give specific referral to ethical approval.

Response: We have provided reference number for the ethical approval.

Comment 8: Lines 192-194: The authors state that they attempted to ensure inclusivity but gained a majority of nursing staff in their sampling. Can this be further discussed?

Response: Nursing staff are the majority of all cadres of HCWs in the hospitals. Hence, higher number of nurses in the study is not surprising.

Comment 9: Is the data for educational level biased as the study authors are comparing very few primary level individuals with tertiary? This holds true for the bivariate analysis as well. Would a doctor or nurse not have a higher education level but also be more likely to accept a vaccine? Please consider further multivariate analyses or further defining the information in the paper.

Response: Thanks for this insightful comment. We conducted a stepwise forwards L-R method which showed that the level of education was the most influential factor rather than the professional categories. Hence, we retain this parameter. Nurses, doctors, pharmacists etc all have tertiary education in comparison to the support staff whom some of them had primary and secondary education.

Comment 10: Lines 270-277: Maybe also good to mention genetic variants of the virus causing issues with vaccine effectiveness and how that can change perception – indeed, this seems to have happened in South Africa due to the February study on AstaZeneca vaccines.

Response: Thanks for this suggestion. We have added more information under the discussion.

Comment 11: The authors mention political sensitivity in their introduction; however, there was no variable to measure how hesitancy or refusal would factor in if governmental trust was compromised. Indeed, this has been a major issue in many countries in the world as they have vaccines available, but the population refuses to be vaccinated due to political distrust (ie: Hong Kong)…please consider touching on this point in the paper. Was any attempt made to measure what type or provenance of the vaccine would influence acceptance? Indeed, this would be a very interesting piece of information whether novel vaccine technology or place of production would influence even highly educated health care workers decisions to be vaccinated.

Response: Indeed the issue of trust in the government as well as the manufacturer of the vaccine or vaccine types and sources are important gaps which need further investigation in future studies. However, these issues were not covered in the current study.

Comment 13: Unfortunately, South Africa has had a very slow vaccination rollout compared to some other countries in their world due to several factors. How many HCWs have actually been vaccinated to date and has their vaccine acceptance been in line with the study results?

Response: Thanks for the comments. We have provided additional information on this issue.

Reviewer 2 Report

Very well written paper on the willingness to be vaccinated with a COVID vaccine by health care workers in South Africa

I have only one minor comment

The title of Table 3. "Binary Logistic regression analysis of demographic and clinical factors associated with vaccine acceptance" does not seem to be appropriate for this Table. 

Demographic actors ok but clinical factors?

Author Response

The title of Table 3. "Binary Logistic regression analysis of demographic and clinical factors associated with vaccine acceptance" does not seem to be appropriate for this Table. Demographic factors ok but clinical factors?

Response: We have made correction on the table.

Reviewer 3 Report

The abstract is only partially informative as It should report that only nurses and support staff was involved. Moreover, It should state that almost all the workers reporter precious contacts with covid patients and they also presented (62 percent) a Major rosk factor for unfavourable covid 19 outcomes such as obesity. The text of the paper and the sections are well organized and presented. However the interest of the readers in this papers could be limited.x

Author Response

The abstract is only partially informative as It should report that only nurses and support staff was involved. Moreover, It should state that almost all the workers reported precious contacts with covid patients and they also presented (62 percent) a major risk factor for unfavourable covid 19 outcomes such as obesity. The text of the paper and the sections are well organized and presented. However the interest of the readers in this papers could be limited.

Response: We have revised the abstract. Thanks